# Development and Reproductive Capacity of the Miyake Spider Mite *Eotetranychus kankitus* (Acari: Tetranychidae) at Different Temperatures

**DOI:** 10.3390/insects13100910

**Published:** 2022-10-07

**Authors:** Mohammad Shaef Ullah, Yurina Kobayashi, Tetsuo Gotoh

**Affiliations:** 1Laboratory of Applied Entomology and Acarology, Department of Entomology, Bangladesh Agricultural University, Mymensingh 2202, Bangladesh; 2Laboratory of Applied Entomology and Zoology, Faculty of Agriculture, Ibaraki University, Ami 300-0393, Ibaraki, Japan; 3Faculty of Economics, Ryutsu Keizai University, Ryugasaki 301-8555, Ibaraki, Japan

**Keywords:** *Eotetranychus kankitus*, bootstrap-match technique, age-stage, two-sex life table, thermal thresholds, distribution

## Abstract

**Simple Summary:**

*Eotetranychus kankitus* (Acari: Tetranychidae) is an important pest of citrus and occurs mainly in Oriental and Palearctic areas. *E. kankitus* often coexists with *Panonychus citri* or *Phyllocoptruta oleivora* (Acari: Tetranychidae, Eriophyidae), resulting in a pest complex that is extremely difficult to control. As adaptable organisms, insects and mites respond differently to environmental changes, especially to changes in temperature that may occur as a result of climate warming. In this study, *E. kankitus* was used as a test organism to see how temperature affects development time, survival rate and fecundity. We used the bootstrap-match technique to construct the life-table parameters. The results showed that oviposition was highest at 20–25 °C, while growth rate was highest at 25–30 °C. Although the optimal temperature for growth differs among spider mite species, the species–specific thermal constants could also be used as a potential indicator of the distribution and abundance of arthropods.

**Abstract:**

*Eotetranychus kankitus* (Acari: Tetranychidae) is an important pest of citrus. Assessing life history parameters is crucial to developing an ecologically sound pest management program. Of the many factors that affect life history parameters of herbivorous insects and mites, temperature has the greatest influence on development rate and reproductive potential. We investigated the effects of temperatures from 15 to 40 °C on the demographic parameters of *E. kankitus* under a long-day (16:8 (L:D) h) photoperiod. The egg-to-adult development time of *E. kankitus* decreased as the temperature increased from 15 to 32.5 °C. At 35 °C, the female laid eggs that died at the larval stage. The estimated lower thermal thresholds (*t*_0_) were 11.01 and 10.48 °C, and the thermal constants (*K*) were 190.67 and 188.63 degree-days for egg-to-adult females and egg-to-adult males, respectively. The intrinsic optimal temperatures (*T_Ø_*) for development were 21.79 and 21.74 °C, respectively. The bootstrap-match technique was used in the construction of the life table paramaters. The net reproductive rate (*R*_0_) decreased as temperature increased from 20 to 30 °C, but the lowest rate was observed at 15 °C. The intrinsic rate of natural increase (*r*) increased from 0.0299 day^−1^ at 15 °C to 0.1822 day^−1^ at 30 °C. These findings provide a critical theoretical basis for predicting the occurrence of *E. kankitus* populations under climate warming and for developing appropriate control strategies.

## 1. Introduction

Most organisms live in an ecosystem that fluctuates within and across generations. Several theoretical and empirical studies have explored the evolution of phenotypic plasticity and acclimation in response to such environmental variability [1,2]. Life processes are closely associated with different abiotic factors. Temperature is one of the most dominant abiotic factors that positively or negatively affects the development and reproduction of poikilothermic arthropods. Temperature affects vital population processes such as development rate, birth rate, death rate, fecundity and generation time, and as a result drives population growth traits of target species [3,4,5,6,7]. Additionally, thermodynamic effects have been observed on biochemical processes and temperature determines physiological functions, which underlie development, fitness, and performance [8].

Life tables include the basic information essential for studying population changes and rates of increase or decrease. Moreover, they can be used to infer the probability of an individual of a certain age surviving and the average number of offspring produced by a female of a given age. Population growth rates largely determine the pest status of spider mites [9], and temperature strongly affects population growth [4,5,10,11]. Many mites, particularly spider mites (Tetranychidae), display significant amounts of life history variation and are prone to temperature-associated differentiation [12,13,14]. Therefore, knowledge regarding the temperature requirements of the different stages of mite pests can be used to forecast their potential distribution and abundance, which can ultimately help to assess the potential severity of a pest species. Overall, life tables are typically constructed by continuously collecting data related to the development, survival, and reproduction of all individuals of a specific cohort from birth to death. However, continuous data collection from birth to death for some insects is inapplicable, impractical, or exceptionally difficult [15]. For example, some insects may have a single generation per year followed by a long dormant period, such as the Sunn pest, *Eurygaster integriceps* Puton (Hemiptera: Scutelleridae), or require very particular conditions for a suitable mating environment (e.g., light intensity, temperature, and humidity) that may be difficult to reproduce in a laboratory setting, such as the biting midge *Forcipomyia taiwana* (Shiraki) (Diptera: Ceratopogonidae) [15]. In these situations, data on immature individuals (developmental time and survival rate) and adults (survival and fecundity) can easily be collected, albeit separately. These situations often cause difficulties in life table analysis and applying the life table data to ecological studies and pest management.

In the present study, the Miyake spider mite *Eotetranychus kankitus* Ehara (Acari: Tetranychidae), an important pest of citrus that occurs mainly in the Oriental and Palearctic areas [16], was used as a test organism. Data were collected on the immature (developmental time and survival rate) and adult mites (survival and fecundity), and a bootstrap-match technique was used to assess the life-table parameters. Although *E. kankitus* does not have as wide a geographic distribution as the citrus red mite *Panonychus citri* (McGregor) (Acari: Tetranychidae) or the citrus rust mite *Phyllocoptruta oleivora* (Ashmead) (Acari: Eriophyidae), it causes much more severe damage where it occurs [17]. In addition, *E. kankitus* can coexist with *P. citri* or *P. oleivora*, which produces a pest complex that is extremely difficult to control in citrus orchards [18,19]. However, it is easy to identify *E. kankitus* and *P. citri* under a stereomicroscope. *Eotetranychus kankitus* is yellowish green in color, and dorsal setae on idiosoma are not on tubercles, while *P. citri* is reddish, and dorsal setae are on strong tubercles.

Ambient temperature is one of the most imperative abiotic factors affecting the survival, development rate, abundance, behavior and fitness of insects and mites. In fact, each insect or mite species has an optimum temperature at which it thrives, with lower and upper limits for development. We hypothesized that high temperatures can decrease fecundity, hatching and survival of these organisms, while low temperatures can affect the sex ratio, behavior and population distribution of insects or mites. To develop an ecologically sound *E. kankitus* pest management system, it is critical to thoroughly understand their life history parameters through life-table studies. Therefore, we used a bootstrap-match technique based on the age-stage, two-sex life table to link the immature data with the adult data and construct a comprehensive life table for *E. kankitus*.

## 2. Materials and Methods

### 2.1. Collection and Rearing of Eotetranychus kankitus

*Eotetranychus kankitus* was originally collected from Sudachi citrus, *Citrus medica* L., at Tokushima Prefecture, Japan (34°03′ N, 134°25′ E) on 11 September 2017. A voucher specimen of *E. kankitus* used in this study is preserved at the Laboratory of Applied Entomology and Zoology, Faculty of Agriculture, Ibaraki University, under the serial voucher specimen number (No. 876). *Eotetranychus kankitus* cultures were reared on leaf discs (16 cm^2^) of Satsuma mandarin, *Citrus reticulata* Blanco, which were placed on water-saturated polyurethane mats in Petri dishes (9 cm diameter, 2 cm depth) and kept in an incubator at 25 ± 0.5 °C and 65 ± 5% RH under a 16:8 (L:D) h photoperiod. The edges of leaf discs prepared with mandarin leaves were soaked with moist tissue paper to prevent mites from escaping. The leaves were replaced after 7–12 days, depending on temperatures and whenever they appeared to be dried out or damaged by feeding *E. kankitus*. The temperature was maintained using the incubators of the same type (MIR-554, Panasonic Co., Osaka, Japan).

### 2.2. Immature E. kankitus Development

To determine the effect of temperature on the development duration of *E. kankitus*, adult females obtained from cultures were transferred individually onto an individual leaf disc (4 cm^2^) and kept at one of 11 constant temperatures from 15 to 40 °C (15, 17.5, 20, 22.5, 25, 27.5, 30, 32.5, 35, 37.5, 40 °C; ± 0.5 °C), under a long-day photoperiod (16:8 (L:D) h) and 65 ± 5% RH for all treatments. Incubators (MIR-554, Panasonic Co., Osaka, Japan) were equipped with temperature control and recording devices, and a water tank was placed in the incubator to control humidity. This species does not enter diapause at all at 15°C under a 8L-16D photoperiod (Gotoh, unpublished data). Females were allowed to lay eggs for 24 h at each test temperature, but eggs laid during this period were discarded to prevent an effect of the 25 °C rearing environment. Subsequently, females were allowed to lay eggs on the leaf for 24 h at temperatures of 15–22.5 °C and 12 h at 25–35 °C; then, the females were removed from the leaf discs. All but one of the eggs produced by a female were destroyed with a needle. Each individual was observed in 12- or 24-h intervals, depending on temperature (24 h at 15–22.5 °C and 12 h at 25–35 °C) and examined with a stereomicroscope (SZ-40, Olympus Corporation, Tokyo, Japan) to record the developmental stage until all individuals reached adulthood.

### 2.3. Development Rate Model

The thermal constant (thermal summation, *K*) of pests and predators is indispensable for estimating the number of generations per year. The thermal constant was calculated using a linear model. Although most models estimate two or three parameters, the only model that can estimate all four (*t*_0_, *K*, *T_Φ_*, and *T_H_*) is Thermodynamic Sharpe-Schoolfield-Ikemoto (SSI) [3,20,21]. Therefore, development rates (calculated as 1/development time) at different constant temperatures were analyzed using this SSI model.

Ikemoto and Takai’s linear model was used to obtain more reliable estimates of the lower thermal threshold and thermal constant [20]. The law of total effective temperature applied to the temperature-dependent development of arthropods is expressed by the following equation:$$\frac{1}{D}=-\frac{t_{0}}{K}+\frac{1}{K}T,$$
where *D*, *T*, *t*_0_, and *K* represent the duration of development (d), environmental (mean/isothermal) temperature (°C), lower thermal threshold and thermal constant, respectively. The equation of the non-linear thermodynamic model was applied for a wide range of temperatures and can be expressed as follows [22,23]:$$D_{r}=\frac{\rho \frac{\left[T\right]}{\left[T_{\Phi }\right]}\exp\left[\frac{\Delta H_{A}}{R}\left(\frac{1}{\left[T_{\Phi }\right]}-\frac{1}{\left[T\right]}\right)\right]}{1+\exp\left[\frac{\Delta H_{L}}{R}\left(\frac{1}{\left[T_{L}\right]}-\frac{1}{\left[T\right]}\right)\right]+\exp\left[\frac{\Delta H_{H}}{R}\left(\frac{1}{\left[T_{H}\right]}-\frac{1}{\left[T\right]}\right)\right]},$$
where *D_r_* represents the development rates (the dependent variables) at the absolute temperature (*T*) (the independent variable). All other parameters are constants: [*T_L_*], [*T_H_*], and [*T_Φ_*] represent the absolute temperatures; Δ*H_A_*, Δ*H_L_*, and Δ*H_H_* represent enthalpy changes; *R* is the universal gas constant; and *ρ* is the developmental rate at [*T_Φ_*]. [*T_Φ_*] is the intrinsic optimum temperature for development that exhibits the minimum effects on enzyme inactivation related to development at low and high temperatures [22], and it is expressed as follows:$$\left[T_{\emptyset }\right]=\frac{\Delta H_{L}-\Delta H_{H}}{R\;\ln\left(-\frac{\Delta H_{L}}{\Delta H_{H}}\right)+\left(\frac{\Delta H_{L}}{\left[T_{L}\right]}\right)-\left(\frac{\Delta H_{H}}{\left[T_{H}\right]}\right)}.$$

The SSI model was modified and developed in the SSI-P program that is run on R statistical software [24]. The SSI model was then improved, and OptimSSI-P was created, in which *T_Φ_* was estimated along with its confidence limits [25].

### 2.4. Reproduction and Adult Longevity

Newly emerged females (0–1-day) obtained from four constant temperature regimens (15, 20, 25, and 30 °C; ± 0.5 °C) were used to assess fecundity and adult longevity. Females were paired with adult males obtained from the same temperature regimens or the mite cultures. If an adult male died earlier than its mate, adult males from the mite cultures were used, but they were not considered for the life table analysis. The leaf discs were observed in 12- or 24-h intervals, depending on temperature (24 h at 15–20 °C and 12 h at 25–30 °C). The number of eggs laid by each female was counted under a stereomicroscope until all mites had died. Eggs laid by individual females were kept until hatching to confirm that they were viable because viable eggs are essential for estimating demographic parameters. Newly emerged males were also observed daily until death to determine their longevity.

### 2.5. Application of the Bootstrap-Match Technique

We used a bootstrap-match technique to match the survival and fecundity data of adult mites with the developmental time and survival data of immature mites [26]. The data of an immature individual was first randomly selected from a cohort of immature mites containing *N_I_* individuals. If the selected immature individual did not complete the immature stage, it was classified as an *N*-type individual. If the selected immature individual had completed all immature stages and emerged as an adult, then the data of an adult of the same sex was randomly selected from the adult cohort of *N_A_* individuals and appended to the immature data to form a complete life history of an individual of one sex. To construct a bootstrap-match cohort totaling *N_S_*, individuals were randomly selected with replacement until a total of *N_S_* individuals were selected. The total number *N_S_* equals *N_I_,* if *N_I_* > *N_A_* and *N_S_* equals *N_A_,* if *N_A_* > *N_I_*. The above procedure was repeated 100,000 times (*B* = 100,000), and a total of 100,000 bootstrap-match life tables were constructed. The bootstrap-match procedure is embedded in the TWOSEX-MSChart program [14].

### 2.6. Demographic Parameters

The life history of individuals at each temperature was analyzed using TWOSEX-MSChart [14]. The age-stage-specific survival rate (*s_xj_*, where *x* = age and *j* = stage), age-specific survival rate (*l_x_*), age-stage-specific fecundity (*f_xj_*), age-specific fecundity (*m_x_*), and population parameters, including net reproductive rate (*R*_0_), intrinsic rate of natural increase (*r*), finite rate of increase (*λ*), and mean generation time (*t*), were calculated using the following equations [27]:$$l_{x}=\sum_{j=1}^{k}s_{xj},$$
where *k* is the number of stages, and
$$m_{x}=\frac{\sum_{j=1}^{k}s_{xj}f_{xj}}{\sum_{j=1}^{k}s_{xj}}\;.$$

The intrinsic rate of natural increase (*r*) was calculated using the Lotka–Euler equation, with age indexed from zero [28], as follows:$$\overset{\infty }{\underset{x=0}{\sum }}e^{-r\left(x+1\right)}l_{x}m_{x}=1.$$

The net reproductive rate (*R*_0_) is the total number of offspring that an individual can produce during its lifetime [28] and was calculated as follows:$$R_{0}=\overset{\infty }{\underset{x=0}{\sum }}l_{x}m_{x}\;.$$

The mean generation time (*t*) represents the amount of time that a population requires to increase its size *R*_0_-fold as time approaches infinity and the population settles to a stable age-stage distribution [28]. Mean generation time was calculated as follows:$$t=\frac{\ln R_{0}}{r}\;.$$

The finite rate of increase (*λ*) is a multiplication factor of the original population at each time period. The finite rate of increase was calculated as follows:$$\lambda =e^{r}\;.$$

The gross reproduction rate (*GRR*) of mites can be used to infer if there was a rapid increase of mite populations that depended on fecundity and adult eclosion, and was calculated as follows:$$GRR=\sum_{x=0}^{\infty }m_{x}\;.$$

The age-stage life expectancy (*e_xj_*) is the time length that an individual of age *x* and stage *j* is expected to survive, and it was calculated as follows:$$e_{xj}=\sum_{i=x}^{\infty }\sum_{y=j}^{k}S_{iy}^{’},$$
where $S_{iy}^{’}$ is the probability that an individual of age *x* and stage *j* will survive to age *i* and stage *y*, and was calculated by assuming $S_{iy}^{’}$ = 1.

Oviposition days (*O_d_*) is the mean number of days on which adult females actually lay eggs and was calculated as follows:$$O_{d}=\frac{\sum_{x=1}^{N_{fr}}D_{x}}{N_{fr}},$$
where *N_fr_* is the number of reproductive females, i.e., females that laid at least one egg, and *D_x_* is the number of oviposition days of the *x*th reproductive female [29].

The mean fecundity of a reproductive female (*F_r_*) was calculated as follows:$$F_{r}=\frac{\sum_{x=1}^{N_{fr}}E_{x}}{N_{fr}},$$
where *E_x_* is the total number of eggs laid by a reproductive female.

The age-stage reproductive value (*v_xj_*) is the contribution of *n* individuals of age *x* and stage *j* to the future population. The reproductive value (*v_xj_*) in the age-stage, two-sex life table was calculated as follows:$$v_{xj}=\frac{e^{r\left(x+1\right)}}{S_{xj}}\overset{\infty }{\underset{i=x}{\sum }}e^{-r\left(i+1\right)}\overset{k}{\underset{y=j}{\sum }}S_{iy}^{’}f_{iy},$$
where $f_{iy}$ is the probability that an individual of age *x* and stage *j* will reproduce to age *i* and stage *y*.

The paired bootstrap test based on the confidence interval of differences was used to assess the differences between life tables [30].

## 3. Results

### 3.1. Immature Mite Development

*Eotetranychus kankitus* females completed their development from egg to adult at temperatures from 15 to 32.5 °C (Table 1). When females laid eggs at 35 °C, some eggs hatched (58.8% hatch rate), but most of them died in the larval stage; at 37.5 °C, few eggs hatched (3.0%), and those that hatched then immediately died in the larval stage; at 40 °C, no eggs hatched (data not shown). Immature survival rates were lower at 15, 17.5, and 32.5 °C than at intermediate temperatures (Table 1).

Egg-to-adult development time decreased as the temperature increased from 15 to 32.5 °C. Development rates for the 15–32.5 °C fit well with Ikemoto and Takai’s linear model (0.9710 ≤ *r*^2^ ≤ 0.9766) (Table 2). The estimated lower thermal thresholds (*t*_0_) for egg-to-adult female and for egg-to-adult male development were 11.01 and 10.48 °C, respectively. The thermal constants (*K*) were 190.67 and 188.63 degree-days for the respective stages (Table 2). The development rate data closely fit Ikemoto and Takai’s non-linear thermodynamic model at temperatures between 15 and 32.5 °C (0.00273 ≤ *χ*^2^ ≤ 0.00336) (Table 2). The intrinsic optimum temperatures (*T_Ø_*) for egg-to-adult female and egg-to-adult male development were 21.79 and 21.74 °C, respectively (Table 2). The upper threshold temperatures (*T_H_*) for egg-to-adult female and egg-to-adult male development were 35.25 and 36.76 °C, respectively (Table 2).

### 3.2. Reproduction and Adult Longevity

The adult preoviposition period and total preoviposition period (days) decreased as the temperature increased from 15 to 30 °C (Table 3). The oviposition period (days) was highest at 20 °C, followed by 25, 15, and 30 °C. Fecundity was highest at 20 °C, followed by 25, 30, and 15 °C (Table 3). Eggs per female per day was increased with the temperature increased from 15 to 30 °C. Adult female and male longevities decreased as the temperature increased from 15 to 30 °C (Table 3).

### 3.3. Age-Stage-Specific Survival Rate

The age-stage-specific survival curve, *s_xj_*, depicts the probability that newly emerged *E. kankitus* individuals will survive to age *x* and stage *j* (Figure 1). For example, when we considered the larval curve at 20 °C (Figure 1), hatching occurred between 6 and 15 days after oviposition. The probability that a larva would survive to the protonymph stage was highest for larvae that hatched nine days after oviposition. The overlaps between different stages (egg, larva, protonymph, deutonymph, and adult) resulted from inter-individual variation in development rates. We found that the curves for the survival of immature and adult mites had the slowest increase at 15 °C and the fastest increase at 30 °C (Figure 1).

### 3.4. Age-Specific Survival and Age-Stage-Specific Fecundity

The age-specific survival rate (*l_x_*) indicated the probability that a newly oviposited egg would survive to age *x*. *l_x_* slowly decreased with time at 15 °C and rapidly decreased at 30 °C (Figure 2). At 30 °C, no individuals survived longer than 25 days (Figure 2). The highest age-stage-specific female fecundity curve (*f_x5_*) showed that 0.64 eggs were produced on the 52nd day, 2.27 eggs on the 29th day, 1.57 eggs on the 21st day, and 1.73 eggs on the 13th day at 15, 20, 25, and 30 °C, respectively (Figure 2). The age-specific fecundity (*m_x_*), defined as the average number of female offspring per female at age *x*, is also shown in Figure 2. Different dynamic patterns for *m_x_* and *l_x_m_x_* were observed at the four temperatures. Reproduction (*m_x_* and *l_x_m_x_*) started earlier with increasing temperatures. The first reproductive peak also occurred earlier with increasing temperatures (Figure 2).

### 3.5. Age-Stage-Specific Life Expectancy

The age-stage-specific life expectancy (*e_xj_*) describes the future expected life span of an individual of age *x* and stage *j* (Figure 3). The life expectancies of newborn mites at 15, 20, 25 and 30 °C were 58.2, 44.7, 27.5 and 18.1 days, respectively (Figure 3), which demonstrated that high temperatures could shorten life expectancies. The *e_xj_* of females fell from 33.9 days at 15 °C to 11.7 days at 30 °C (Figure 3).

### 3.6. Age-Stage-Specific Reproductive Value

Reproductive value is defined as an individual’s contribution to the future population. The age-stage-specific reproductive values (*v_xj_*) of female *E. kankitus* reared under four constant temperatures at age *x* and stage *j* are presented in Figure 4. The effect of age on the reproductive values could be clearly observed in the *v_xj_* curves. The peak *v_xj_* of adult females at 15, 20, 25 and 30 °C were 7.1, 16.7, 13.3 and 9.9 day^−1^, respectively (Figure 4); this indicated that high temperature (25 and 30 °C) could shorten reproductive value.

### 3.7. Population Parameters

The net reproductive rate (*R*_0_) generally decreased as temperature increased from 20 to 30 °C, but the lowest *R*_0_ was observed at 15 °C (*p* < 0.05; Table 4). The intrinsic rate of natural increase (*r*) increased from 0.0299 day^−1^ at 15 °C to 0.1822 day^−1^ at 30 °C (Table 4). The finite rate of increase (λ) changed with increasing temperature from 15 to 30 °C. The mean generation time (*t*) decreased from 55.95 days at 15 °C to 14.61 days at 30 °C (Table 4). The gross reproductive rate also showed a similar trend to the net reproductive rate and was highest at 25 and 20 °C (Table 4).

## 4. Discussion

Life tables are the most suitable tools for studying insect and mite populations and play a crucial role in biology and ecology [12,13,14]. In addition, knowledge about the adaptation of insects and mites to climatic conditions plays a vital role in predicting growth times, reproduction, hibernation and distribution and therefore can be used to develop suitable pest management programs [31]. In this study, temperature significantly affected the development, reproduction and demographic parameters of *E. kankitus*. Our results demonstrate that *E. kankitus* could successfully develop from egg to adult and produce offspring at temperatures ranging from 15 to 32.5 °C. However, temperatures at 15 and 30 °C had a detrimental effect on fecundity, and this vulnerability to low and high temperatures restricted its distribution [16].

In many temperature-dependent development studies of arthropods, linear models were used to estimate the lower thermal threshold and thermal constant [12,32]. An insect’s lower thermal threshold and thermal constant are indicators that can be used to forecast its potential distribution and abundance [3,6,7,33,34]. Several equations have been developed to describe the relationship between development rate and temperature. The degree-day method, based on the number of degree-days required to complete development, is generally used to estimate physiological time. It has several advantages, such as simplicity and the ability to easily calculate the thermal constant [35]. Equations given by Ikemoto and Takai and Ikemoto et al. can more precisely estimate the lower thermal threshold, thermal constant, optimum temperature, and upper threshold temperature [20,25]. The development rate of *E. kankitus* increased almost linearly with an increase in temperature to 32.5 °C. An excellent fit was obtained by Ikemoto and Takai’s linear model for egg-to-adult (female and male) development of *E. kankitus* [20], which is based on the obtained coefficient of determination (0.9710 ≤ *r*^2^ ≤ 0.9766). The estimated lower thermal thresholds (*t*_0_) of *E. kankitus* were 11 and 10.5 °C, and thermal constants (*K*) were 190.7 and 188.6 degree-days for egg-to-adult female and egg-to-adult male development, respectively (Table 2). These values for the lower thermal threshold of *E. kankitus* indicate that they can complete their development at lower temperatures.

Non-linear models require more effort to both find the best fit and interpret the role of the independent variables. It is also essential to choose initial starting points to avoid convergence to the optimum temperature [36]. According to the nonlinear thermodynamic model, the optimum developmental temperature for *E. kankitus* was 21.7–21.8 °C. In our study, the immature survival rate was lowest at 15 and 32.5 °C; the optimal temperature for juvenile survival occurred at the midpoint of this range. The upper (*T_H_*) threshold temperatures were estimated at 35.2 and 36.9 °C for egg-to-adult females and egg-to-adult males, respectively, suggesting that the hypothetical enzyme was half active and half inactive at these thresholds. This could be one of the explanations for the incomplete development of the egg and larval stage at >35 °C. The intrinsic optimum temperature and its confidence interval could be used as an indicator of the geographical distribution and place of origin of related species [22,23].

In a previous study, it was observed that the mean adult longevities of females and males of *E. kankitus* were 12.7 and 8.7 days, respectively, at 28 °C and 80% RH [37]. The maximum age-specific mean fecundity of females was 6.2 eggs at 12 days, and the maximum lifetime fecundity of females was 123 eggs. In addition, females from 8 to 22 days old had the highest fecundity. The means, variances and standard errors of population parameters of *E. kankitus* were evaluated using a bootstrap-match technique with 10,000 replications. The intrinsic rate of natural increase (*r*), finite rate of increase (*λ*), net reproduction rate (*R*_0_), mean generation time (*t*) and gross reproduction rate (*GRR*) of *E. kankitus* were 0.2607 day^−1^, 1.2978 day^−1^, 28.55 offspring, 12.86 days and 49.86 offspring, respectively [37]. Our study demonstrated significant differences in life-history parameters of *E. kankitus* populations compared with the findings of Li et al. [37]. The possible reasons for variation between the two studies include differences in geographical populations, temperature, humidity, photoperiod and rearing conditions, which may impact the changes in development and reproductive parameters and therefore the life-table parameters.

Knowledge of the temperature effects on the intrinsic rate of natural increase (*r*) and finite rate of increase (*λ*) of pest populations permits a comprehensive assessment of outbreaks. The relationship among *R*_0_, *r* and *λ* was shown as: if *R*_0_ > 1, then *r* > 0 and *λ* > 1; if *R*_0_ = 1, then *r* = 0 and λ = 1; if *R*_0_ < 1, then *r* < 0 and λ < 1 [38,39]. It was confirmed that the *r* value could substantially increase if the initial reproduction began one day earlier [40]. All net reproductive rates (*R*_0_) and mean fecundities (*F*) observed in this study were consistent with the Chi’s proof, i.e., *R*_0_ = *F* × *N_f_*/*N* [41]. The results of this study are all consistent with the previous findings described above. Small increases in *r* values can result in significantly faster population growth because of the exponential nature of *r* in the Euler–Lotka equation. In the present study, the *r* values of *E. kankitus* increased as the temperature increased from 15 to 30 °C. However, the temperature dependence of fecundity or offspring production alone cannot accurately reveal the overall effect of temperature on population fitness [42]. For example, at 32.5 °C, development rates and the intrinsic rate of natural increase were higher despite the very low fecundity. Thus, a life table offers a more comprehensive evaluation of population fitness than development rate or fecundity alone.

An innovative bootstrap-match technique was used to construct reliable and complete life tables. However, this technique only considers the following situations: (1) immature and adult life table data were collected separately because of obligatory diapause of one of the life stages; (2) the immature and adult life table data were collected separately because of difficulties in mating of adults under controlled laboratory conditions; (3) a large cohort was used to collect data for immature individuals, but a much smaller cohort was used for the adult data; or (4) the life table data for immature individuals were collected using individually reared insects or mites, whereas the adult data were obtained using group-reared insects [26]. In this study, *E. kankitus* obtained a large cohort of immature individuals’ data, but a smaller cohort for the adult data.

It is vital to construct a precise predictive model for adult emergence and a forecast strategy that can assist as a critical component of an integrated pest management system wherein it enables decision making and improves control efficacy. Predicting the precise seasonal occurrence of agricultural insect/mite pests, including *E. kankitus,* is significant for scheduling, sampling, and the selection of control tactics. Climatic factors in general have been shown to play substantial roles in insect life; among such climatic factors, temperature has the utmost influence on population dynamics and the timing of biological events of insect and mite species. The development of mites occurs within narrow temperature ranges that vary between different mite species and is sensitive to temperature changes. There are still substantive uncertainties to be considered, including uncertainty as to how generalizable the results are to open field conditions. The survival rates, development rates, reproductive capacities and population growth of spider mites under fluctuating temperatures are disparate from those under constant temperatures [13,43,44]. Henceforth, fluctuating temperatures should be taken into account to predict the population dynamics of spider mites in nature. Climate change affects the pattern of population dynamics of insects and mites in different ways. Global warming not only leads to greater over-winter survival, earlier appearance in spring, an increase in the number of generations in a year, lengthening of the reproductive season, etc., but also affects their biotic associations as a result of changes in interspecific interactions. Both the temperature threshold and the maximum temperature using the SSI nonlinear model were employed to estimate the temperature ranges for mite development, because the model provides clear biophysical meaning and thermodynamic information among model parameters. More precisely, comprehensive knowledge of thermal effects on arthropods can be used to determine the range where the pest might develop or establish and to forecast the population’s growth potential and dynamics. A precise estimation of the pest’s population parameters is essential to develop a pest management program [45].

## 5. Conclusions

Our results highlight the need to investigate the impact of temperature on the life-history variation and population growth rate of spider mites. We found that median temperatures (20–25 °C) were favorable for oviposition, but the growth rate potential of *E. kankitus* was higher at 25–30 °C in a stable environment. Although the temperature requirements for a higher growth rate of different spider mite species differed, the species-specific thermal constants could be used as a potential indicator of the distribution and abundance of these arthropods.

## Figures and Tables

**Figure 1 insects-13-00910-f001:**
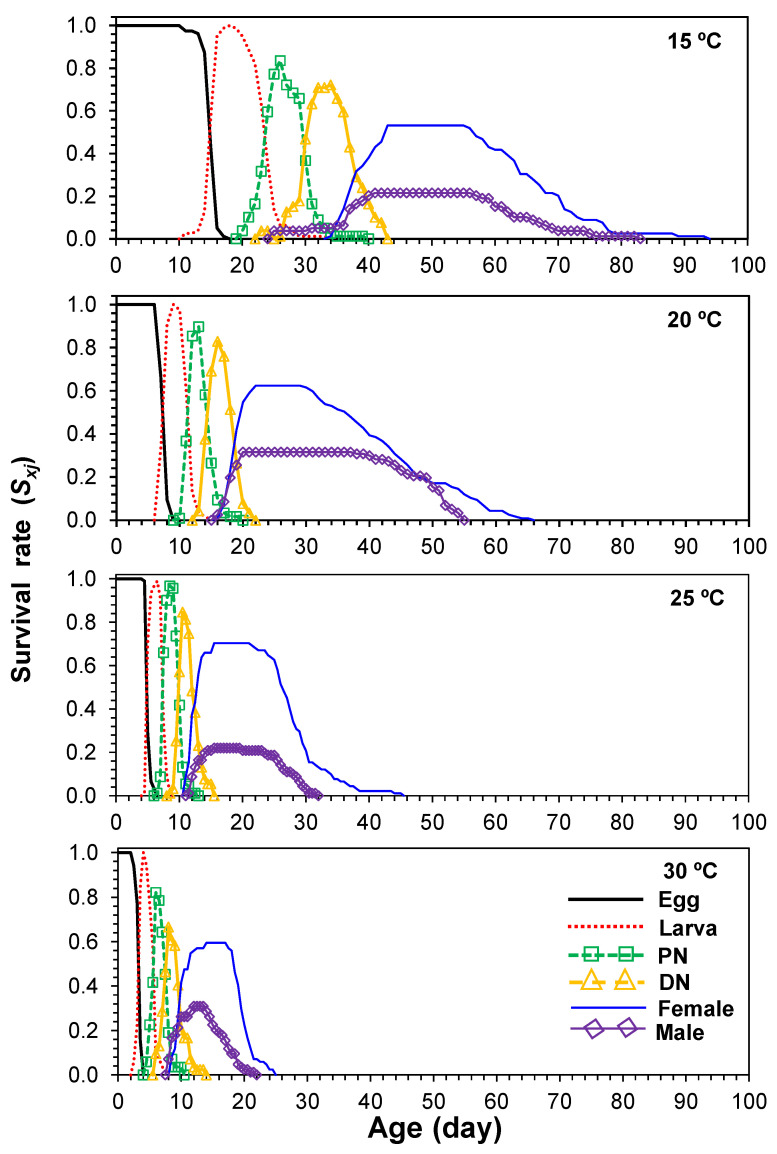
Age-stage-specific survival rates (*s_xj_*) of *Eotetranychus kankitus* at different temperatures (PN = protonymph and DN = deutonymph).

**Figure 2 insects-13-00910-f002:**
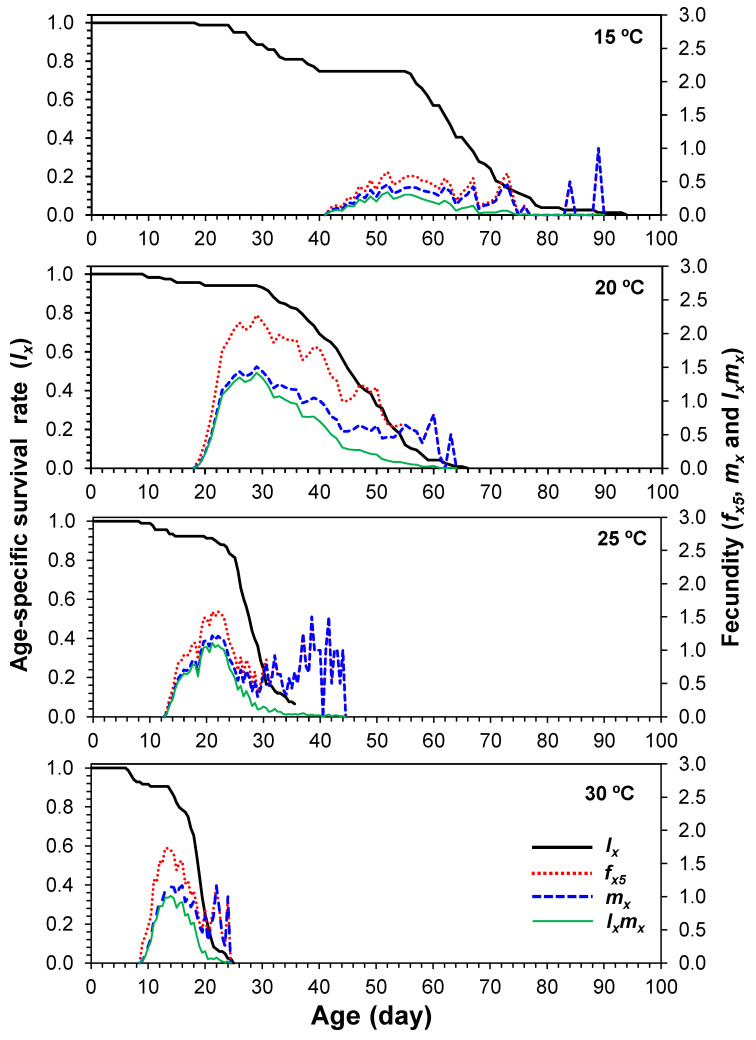
Age-specific survival (*l_x_*), age-stage-specific fecundity (*f_x5_*), age-specific fecundity (*m_x_*), and age-specific maternity (*l_x_m_x_*) of *Eotetranychus kankitus* at different temperatures.

**Figure 3 insects-13-00910-f003:**
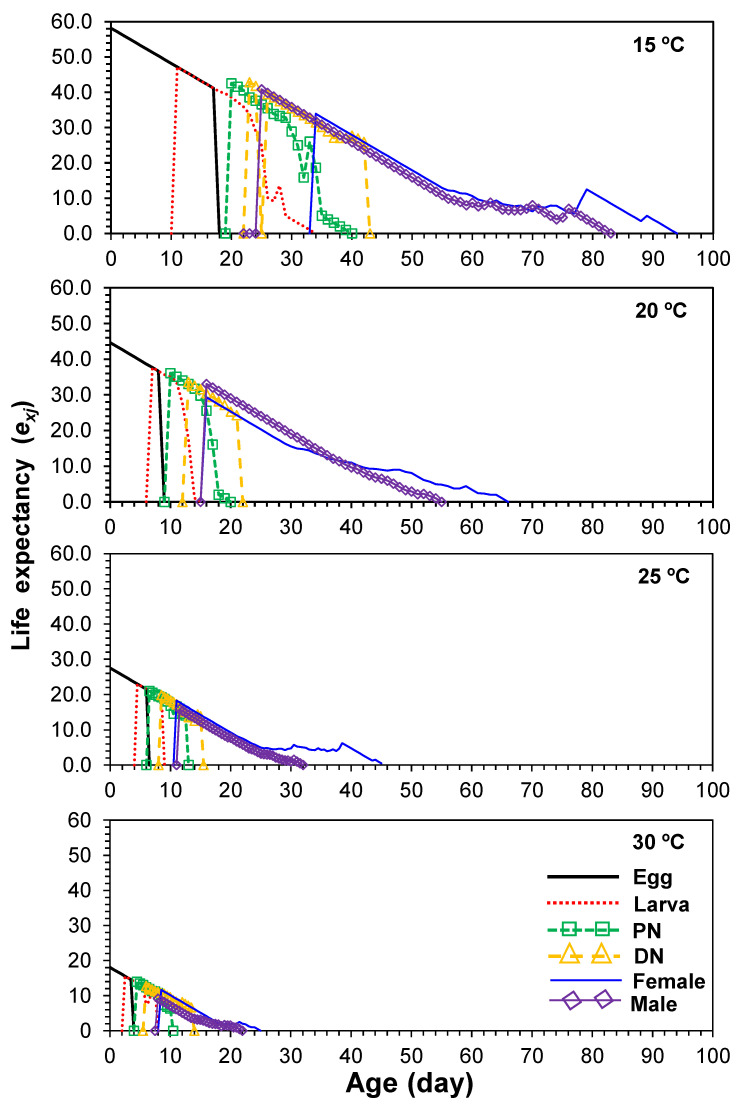
Age-stage-specific life expectancy (*e_xj_*) of *Eotetranychus kankitus* at different temperatures (PN = protonymph and DN = deutonymph).

**Figure 4 insects-13-00910-f004:**
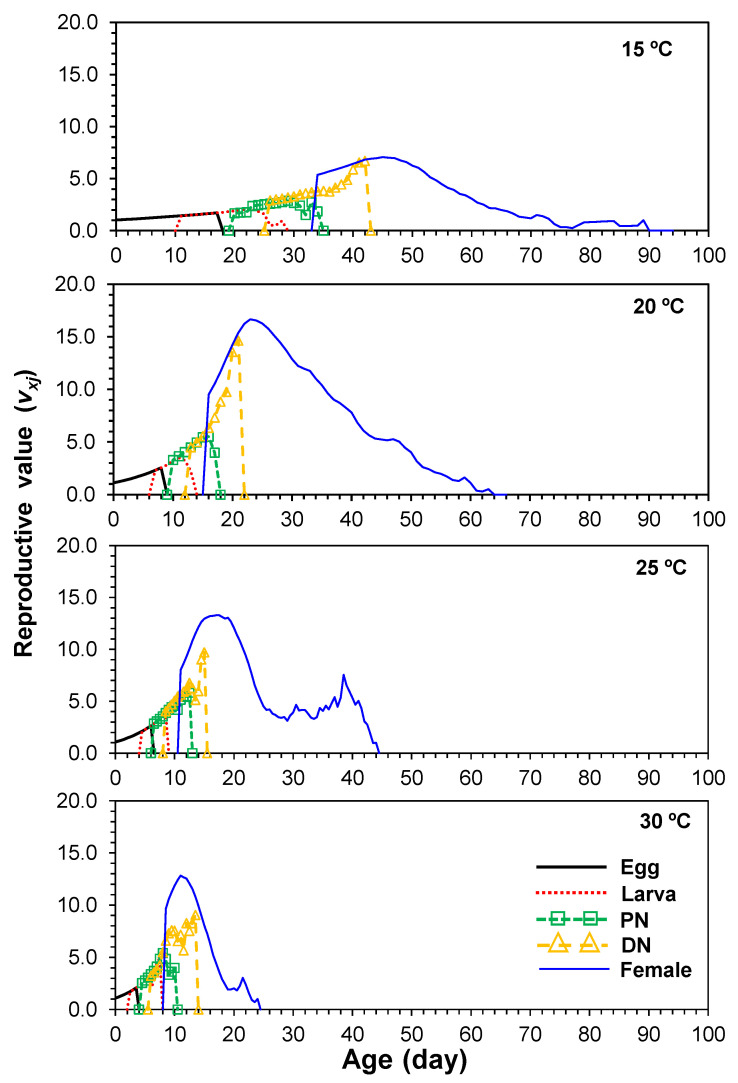
Age-stage-specific fecundity (*v_xj_*) of *Eotetranychus kankitus* at different temperatures (PN = protonymph and DN = deutonymph).

**Table 1 insects-13-00910-t001:** Development duration (days ± S.E.) from egg to adult of *Eotetranychus kankitus* reared on mandarin leaves at different temperatures under a 16L:8D photoperiod.

Temperature	Sex	*n* ^a^	Egg	Larva	Protonymph	Deutonymph	Egg to Adult	% Survival	% Females
15 °C	♀	42	15.36 ± 0.11 b	8.17 ± 0.20 a	6.43 ± 0.25 b	8.57 ± 0.28 b	38.52 ± 0.42 a	74.7 b	71.2
	♂	17	15.76 ± 0.16 a	7.53 ± 0.36 d	5.35 ± 0.36 d	6.59 ± 0.61 d	35.24 ± 1.30 b		
17.5 °C	♀	38	10.63 ± 0.09 c	8.11 ± 0.22 b	7.29 ± 0.18 a	8.71 ± 0.20 a	34.74± 0.38 c	69.5 b	66.7
	♂	19	10.21 ± 0.25 d	7.79 ± 0.32 c	6.11 ± 0.27 c	7.42 ± 0.35 c	31.53 ± 0.79 d		
20 °C	♀	73	7.73 ± 0.08 f	4.00 ± 0.09 e	3.30 ± 0.08 e	3.97 ± 0.08 e	19.00 ± 0.16 e	94.0 a	66.4
	♂	37	7.86 ± 0.09 e	3.76 ± 0.11 f	3.08 ± 0.10 f	3.51 ± 0.10 f	18.22 ± 0.20 f		
22.5 °C	♀	69	5.55 ± 0.06 h	3.77 ± 0.08 f	2.90 ± 0.09 g	3.97 ± 0.12 e	16.19 ± 0.22 g	92.5 a	69.7
	♂	30	5.93 ± 0.08 g	3.63 ± 0.15 g	2.50 ± 0.11 h	3.33 ± 0.13 g	15.40 ± 0.29 h		
25 °C	♀	64	5.15 ± 0.05 j	2.52 ± 0.05 j	2.49 ± 0.05 h	2.34 ± 0.08 j	12.50 ± 0.14 j	92.3 a	76.2
	♂	20	5.28 ± 0.07 i	2.38 ± 0.05 k	2.42 ± 0.08 i	2.75 ± 0.18 h	12.82 ± 0.24 i		
27.5 °C	♀	66	4.23 ± 0.05 l	3.21 ± 0.06 h	2.25 ± 0.06 j	2.74 ± 0.08 h	12.44 ± 0.15 j	87.1 a	81.5
	♂	15	4.37 ± 0.08 k	3.10 ± 0.15 i	2.10 ± 0.11 k	2.53 ± 0.10 i	12.10 ± 0.18 k		
30 °C	♀	50	3.52 ± 0.04 m	2.36 ± 0.06 k	1.98 ± 0.06 l	2.23 ± 0.09 k	10.09 ± 0.18 l	90.5 a	65.8
	♂	26	3.17 ± 0.07 p	2.13 ± 0.07 l	1.88 ± 0.12 m	2.06 ± 0.10 m	9.25 ± 0.25 n		
32.5 °C	♀	51	3.45 ± 0.02 o	1.81 ± 0.04 m	1.58 ± 0.06 n	2.55 ± 0.11 i	9.39 ± 0.16 m	65.7 b	76.1
	♂	16	3.50 ± 0.05 n	1.75 ± 0.06 n	1.53 ± 0.09 o	2.16 ± 0.10 l	8.94 ± 0.14 o		

^a^ Number of individuals tested; means in the same column followed by different letters denote significant differences in different temperatures based on the paired bootstrap test at 5% significance level.

**Table 2 insects-13-00910-t002:** Estimated values of constants in linear and non-linear thermodynamic model describing the relationship between temperature (°C) and developmental rates in *Eotetranychus kankitus* on mandarin leaves.

Model	Parameter	Stage	
		Egg-to-Female Adult	Egg-to-Male Adult
Ikemoto and Takai linear model (Ikemoto and Takai 2000)	*t_0_ *^a^ (S. E.)	11.01 ± 1.72	10.48 ± 1.83
	*K*^b^ (S. E.)	190.67 ± 32.76	188.63 ± 32.32
	*y =*	0.0052x − 0.0578	0.0053x − 0.0555
	*r* ^2^	0.9710	0.9766
SSI (Ikemoto, 2005; 2008) ^c^	*T_Φ_* (K) ^d^	295.05	294.89
	*T_Φ_* (°C)	21.79	21.74
	𝜌	0.05711	0.05974
	Δ*H_A_* (cal/mol)	15,298	14,748
	Δ*H_L_* (cal/deg.mol)	−72,587	−67,035
	Δ*H_H_* (cal/deg.mol)	54,325	48,833
	*T_L_ *(K) ^d^	284.72	284.08
	*T_H_ *(K) ^d^	308.40	309.91
	𝜒 ^2^	0.00336	0.00273

^a^ *t*_0_: lower thermal threshold (°C); ^b^ *K*: thermal constant (°C); ^c^ *T_Φ_*: intrinsic optimum temperature; 𝜌: developmental rate at *T_Φ_*; Δ*H_A_*: enthalpy of activation of the reaction that is catalyzed by the enzyme; Δ*H_L_* and Δ*H**_H_*: change in enthalpy associated with low and high temperature inactivation of the enzyme, respectively; *T_L_* and *T_H_*: temperature at which the enzyme is half active and half low and high temperature inactive, respectively; ^d^ *K*: unit of thermodynamic temperature at which 273.15 K equals 0 °C.

**Table 3 insects-13-00910-t003:** Average pre-oviposition period (APOP), total pre-oviposition period (TPOP), oviposition days, female and male longevity (days ± S.E.), and eggs per female (mean ± S.E.) of *Eotetranychus kankitus* reared on mandarin leaves at different temperatures under a 16L:8D photoperiod.

Temperature	*n* ^a^	APOP	TPOP	Oviposition Days	Female Longevity	Male Longevity	Eggs per Female/Day	Eggs per Female
15 °C	42	10.74 ± 0.56 a	49.26 ± 0.69 a	8.69 ± 0.70 c	67.90 ± 1.30 a	65.82 ± 1.72 a	0.7 ± 0.04 d	10.02 ± 0.60 d
20 °C	73	2.82 ± 0.06 b	21.82 ± 0.17 b	20.67 ± 0.87 a	45.30 ± 1.11 b	49.08 ± 0.77 b	1.9 ± 0.06 c	40.27 ± 1.78 a
25 °C	64	2.14 ± 0.05 c	14.64 ± 0.15 c	10.80 ± 0.45 b	29.24 ± 0.58 c	27.35 ± 0.64 c	2.3 ± 0.12 b	31.28 ± 1.41 b
30 °C	50	0.84 ± 0.03 d	10.93 ± 0.18 d	7.65 ± 0.17 d	20.16 ± 0.25 d	17.17 ± 0.43 d	2.9 ± 0.12 a	24.06 ± 0.90 c

^a^ Number of individuals tested; means in the same column followed by different letters denote significant differences based on the paired bootstrap test at 5% significance level.

**Table 4 insects-13-00910-t004:** Demographic parameters (mean ± S. E.) of *Eotetranychus kankitus* reared on mandarin leaves at different temperatures under a 16L:8D photoperiod. Net reproductive rate (*R*_0_), intrinsic rate of natural increase (*r*, day^−1^), mean generation time (*t*, day), finite rate of increase (*λ*), and gross reproduction rate (*GRR*).

Temperature	*R* _0_	*r*	*t*	*λ*	GRR
15 °C	5.33 ± 0.65 c	0.0299 ± 0.0023 d	55.95 ± 0.77 a	1.0304 ± 0.0023 d	11.09 ± 1.15 c
20 °C	25.13 ± 2.11 a	0.1041 ± 0.0028 c	30.96 ± 0.35 b	1.1098 ± 0.0030 c	36.52 ± 2.87 a
25 °C	22.00 ± 1.80 a	0.1493 ± 0.0041 b	20.70 ± 0.23 c	1.1610 ± 0.0048 b	46.93 ± 6.08 a
30 °C	14.32 ± 1.40 b	0.1822 ± 0.0073 a	14.61 ± 0.20 d	1.1998 ± 0.0087 a	23.19 ± 1.83 b

Means in the same column followed by different letters denote significant differences based on the paired bootstrap test at 5% significance level.

## Data Availability

The datasets generated in this study are available from the corresponding author on reasonable request.

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
