# Peer review of "Development and Reproductive Capacity of the Miyake Spider Mite *Eotetranychus kankitus* (Acari: Tetranychidae) at Different Temperatures"

_insects, 2022, doi:10.3390/insects13100910_

Round 1
Reviewer 1 Report
Dear Colleagues,
The paper is interesting and well written. My biggest concern is about the identifcation of the mite species used in the study, Eotetranychus Kankitus.
The authors say in the paper that another species of spider mite occurs in the same host plant, Panonychus citri.
I don't know if it is possible to differentiate these two species under a stereomicroscope, but it would be interesting to include in the paper how the mites were identify.
I also consider important to have vouchers of the mites used in the study mounted in slides and deposited in a collection. This may be useful in future studies.
Please, consider including these information in the paper as it may avoid possible errors in the identification of mites and in future studies.
Best regards.
Author Response
#1
We greatly appreciate the reviewers’ comments. We made several changes to the manuscript carefully considering all comments and questions raised by the reviewers. Our point-by-point responses are listed below (in red). The changes in the manuscript are shown in track changes mode. We hope that our improved manuscript will meet your approval.
Dear Colleagues,
The paper is interesting and well written. My biggest concern is about the identification of the mite species used in the study, Eotetranychus Kankitus.
The authors say in the paper that another species of spider mite occurs in the same host plant, Panonychus citri.
I don't know if it is possible to differentiate these two species under a stereomicroscope, but it would be interesting to include in the paper how the mites were identify. ---> It’s easy to identify them under a stereomicroscope. We added the following sentence in the text. “However, it’s easy to identify E. kankitus and P. citri under a stereomicroscope. Eotetranychus kankitus is yellowish green in color and dorsal setae on idiosoma are not on tubercles, while P. citri is reddish and dorsal setae are on strong tubercles.”
I also consider important to have vouchers of the mites used in the study mounted in slides and deposited in a collection. This may be useful in future studies. ---> Yes, it is very important and we added this information as: “Voucher specimen of E. kankitus used in this study is preserved at the Laboratory of Applied Entomology and Zoology, Faculty of Agriculture, Ibaraki University, under the serial voucher specimen number (no. 876).”
Please, consider including these information in the paper as it may avoid possible errors in the identification of mites and in future studies. . ---> We incorporated these information in the revised draft. Thank you very much.
Reviewer 2 Report
Please find my comments in the attached document.

Author Response
#2
We greatly appreciate the reviewers’ comments. We made several changes to the manuscript carefully considering all comments and questions raised by the reviewers. Our point-by-point responses are listed below (in red). The changes in the manuscript are shown in track changes mode. We hope that our improved manuscript will meet your approval.
General remarks
The manuscript (insects-1932806) by Shaef Ullah et al., presents a detailed life table for the spider mite Eotetranychus kankitus. The authors were very methodic with the data collection and presentation. The manuscript, however, can be greatly improved to present a broader overview of the study and the pest itself. It seems that the authors are presenting justifications as to why they chose to generate another life table for the same mite species. Li et al. 2017 has already published this information. Hence, I recommend that the authors highlight the importance of their study within an Integrated Pest Management framework. ---> Li et al. (2017) studied the life table of E. kankitus only at 28 ºC, and we construct the life table data for temperatures from 15 to 40°C. In addition, we measured the lower thermal threshold and thermal constant to know the potential distribution of this pest. We discuss the findings of Li et al (2017) in the Discussion section.
Added ‘More precisely, comprehensive knowledge of thermal effects on arthropods can be used to determine the range where the pest might develop or establish and to forecast the population’s growth potential and dynamics. Actually, a precise estimation of the pest’s population parameters is essential to develop a pest management programme [43].’ In Discussion.
Zamani, A.A., A.A. Talebi, Y. Fathipour, and V. Baniameri. 2006. Effect of temperature on biology and population growth parameters of Aphis gossypii Glover (Hom., Aphididae) on greenhouse cucumber. J. Appl. Entomol. 130: 453– 460.
The introduction can be enhanced to present literature related to the pest, its economic importance, distribution, host range and explain how the study is going to add information in this body of literature.
>There is a very little information published on Eotetranychus kankitus and we already added it in the Introduction section:
In the present study, Eotetranychus kankitus Ehara (Acari: Tetranychidae), one of the most important pests of citrus that occurs mainly in the Oriental and Palearctic areas [16]. Although E. kankitus does not have as wide a geographic distribution as the citrus red mite Panonychus citri (McGregor) (Acari: Tetranychidae) or the citrus rust mite Phyllocoptruta oleivora (Ashmead) (Acari: Eriophyidae), it causes much more severe damage where it occurs [17]. In addition, E. kankitus can coexist with P. citri or P. oleivora, which produces a pest complex that is extremely difficult to control in citrus orchards [18, 19].
Similarly, in the discussion the authors fail to present their results and their significance in a broader context. Since this is a pest, the authors can present the current management practices and how can be adjusted based on the biology data that were presented.
>See ‘Specific questions’ section.
I was also wondering whether the authors can provide data on the sex ratio.
> Added the F1 Sex ratio data in Table 1.
Specific questions that were not answered either in the introduction or discussion are:
- Why do you study this mite?
- Eotetranychus kankitus Ehara (Acari: Tetranychidae), one of the most important pests of citrus that occurs mainly in the Oriental and Palearctic areas including Japan.
- How will you use the generated information?
- Added ‘More precisely, comprehensive knowledge of thermal effects on arthropods can be used to determine the range where the pest might develop or establish and to forecast the population’s growth potential and dynamics. Actually, a precise estimation of the pest’s population parameters is essential to develop a pest management programme [43].’ In Discussion.
- What is the significance of your findings?
- Our findings can help integrate the pest management program considering the pest biology at a given temperature.
- How the different thermodynamic models presented in the current study can be correlated with the global warming literature?
- Added ‘Climate change affects the pattern of population dynamics of insects and mites in different ways. Global warming not only leads to greater over-winter survival, earlier appearance in spring, an increase in the number of generations in a year, lengthening of the reproductive season, etc., but also affects their biotic associations as a result of changes in interspecific interactions. Both the temperature threshold and the maximum temperature using the SSI nonlinear model were employed to estimate the temperature ranges for mite development, because the model provides clear biophysical meaning and thermodynamic information among model parameters.’ in Discussion
- Given the fact that in nature multiple pests can infest a plant at the same time, what other competitors will the E. kankitus face?
> Yes, multiple pests can infest in a plant at the same time, mentioned in Introduction section. No other competitors are recorded yet, so our study focused on the temperature effects on biological parameters on E. kankitus.
Specific and/or minor remarks
L. 47 affects the development
> Done
L. 46-50 This is a long sentence and a bit difficult to follow, please consider revising.
> Revised
- 58 remove acarine, which is synonym to mites
> Removed
- 64-80 This paragraph appears to be repetitive and does not add useful information. Please consider merge it with the previous and instead add more information about the pest (see also general remarks).
> Revised
- 99 add coordinates
> (34°03’N, 134°25’E)
L. 104-105 please describe the method used to rear the pest with a bit more details
> We added ‘The temperature was maintained using the incubators of the same type (MIR-554, Panasonic Co., Osaka, Japan).’
- 111 a leaf disc of which plant species?
> added ‘of citrus’
- 112 what was the RH?
> added ‘65 ± 5% RH for all treatments. Incubators (MIR-554, Panasonic Co., Osaka, Japan) were equipped with temperature control and recording devices, and a water tank was placed in the incubator to control humidity.’
L.151 how old exactly?
> added ‘0-1-days’
- 230 eggs did not hatch, or no eggs hatched
> Replaced ‘all eggs died’ by ‘no eggs hatched’
- 235 Italic in the Eotetranychus kankitus, this comment is applicable to all table and figure legends of the manuscript
> Done
- 272 Can the authors add to this table the mean number of eggs per female per day?
> Added in the Table 3
- 290 survival not survivability, please make the correction throughout the manuscript
> Done
- 409 this important information of the diapausing stage was not presented earlier and the authors do not elaborate. Please add more information in the introduction about the diapause of this species and discuss your results within this context.
> There is no report so far on the diapause of the present species. We added it in M & M as follows: This species does not enter diapause at all at 15°C under a 8L-16D photoperiod (Gotoh, unpublished data).
- 414 why insects are mentioned here?
> revised is as ‘insects or mites’
L. 415-416 Please remove this sentence, it has already be presented in L. 411-413
> replaced it with ‘E. kankitus obtained large cohort of immature individuals data, but smaller cohort for the adult data.’
Reviewer 3 Report
The authors have graphed and presented their results clearly, drawing some attention to the implications of their findings. I found the study of interest and a good contribution to the knowledge of bioecology of mites. The methods used are appropriate for the objectives of the work and, in general, well depicted. The resulting figures are sufficient, informative, and of good quality helping to follow the reasoning throughout the manuscript. Nevertheless, the manuscript would benefit from a more thorough literature review in my humble opinion.
Firstly, the Intro and Discussion provide no insight on how this MS relates to the various other ones cited in the text or concerns that have been raised by other researchers. The authors do not present any hypotheses or expectations that could be connected to previous studies; adding these details will improve the paper. The authors should clearly explain why the study was done, why it was important, and how it fits with other studies. It should be clear and concise. The intro should also include what outcome(s) they expect, and how it would help support or refute their hypotheses or answer their questions.
Also, authors should explain how they have arrived to using solely the SSI model for their development predictions, i.e., the estimated thresholds may not quite equate to the biological threshold above which this mite develops. Studies on the effects of temperature on insect and mite development have been criticized because analyses commonly use models that are considered standard to the field of investigation or are preferred for a particular taxonomic group (see https://doi.org/10.1093/aesa/saw098, or https://doi.org/10.1093/aesa/sax063). As a result, alternative models that could provide superior fits to experimental datasets may be overlooked. Other important criteria that should be considered for fitting nonlinear models to temperature-driven development rate data for insects include use of parameters that have biological relevance are close-to-linear, which means that the least squares estimators are close to being normally distributed and are unbiased, minimum variance estimators. In these situations, good initial parameter estimates can be obtained which promotes successful model convergence (see https://doi.org/10.1093/jee/toz320).
My final concern is that the authors are extrapolating the applicability of their results beyond what the design supports. These are only development data from sets of highly artificial constant temperatures, so the inference power of the paper is very limited, but authors do not acknowledge this detail at all and need to be more forthcoming. This is a critical limitation of the study, and the authors must concede and discuss this. It is well known that most of laboratory experiments are conducted under constant temperatures whereas in nature daily temperature fluctuations can be very wide. The interaction of cyclic temperatures with nonlinear development or life history parameters can introduce significant deviations from the parameters developed here. So, I am suggesting to the authors to tone-down the language a little and admit that there are still substantive uncertainties to be considered, including uncertainty as to how generalizable the results are to open field conditions. Some of the authors statement would for example be much stronger if they tie and compare their work to the body of literature that has built up from rearing insects and mites at a wide set of realistic fluctuating temperatures. This is not to diminish the data gathered in this study, they are of value. But it is important for the authors not to overgeneralize, and to warn the reader against doing so as well. Some suggestions might include but are not limited to J. Econ. Entomol. 113: 633-645., J. Econ. Entomol. 112: 1560-1574., J. Econ. Entomol. 112:1062-1072., etc. etc. Adding these details will improve the paper in my opinion.
Good luck!
Author Response
#3
We greatly appreciate the reviewers’ comments. We made several changes to the manuscript carefully considering all comments and questions raised by the reviewers. Our point-by-point responses are listed below (in red). The changes in the manuscript are shown in track changes mode. We hope that our improved manuscript will meet your approval.
The authors have graphed and presented their results clearly, drawing some attention to the implications of their findings. I found the study of interest and a good contribution to the knowledge of bioecology of mites. The methods used are appropriate for the objectives of the work and, in general, well depicted. The resulting figures are sufficient, informative, and of good quality helping to follow the reasoning throughout the manuscript. Nevertheless, the manuscript would benefit from a more thorough literature review in my humble opinion.
Firstly, the Intro and Discussion provide no insight on how this MS relates to the various other ones cited in the text or concerns that have been raised by other researchers. The authors do not present any hypotheses or expectations that could be connected to previous studies; adding these details will improve the paper. The authors should clearly explain why the study was done, why it was important, and how it fits with other studies. It should be clear and concise. The intro should also include what outcome(s) they expect, and how it would help support or refute their hypotheses or answer their questions.
>Added ‘Ambient temperature is one of the most imperative abiotic factors affecting the survival, development rate, abundance, behavior and fitness of insects and mites. In fact, each insect or mite species has an optimum temperature at which they thrive, with lower and upper limits for development. We hypothesized that high temperatures can decrease fecundity, hatching and survival of these organisms, while low temperatures can affect the sex ratio, behavior, and population distribution of insects or mites.’ in Introduction
>Added ‘More precisely, comprehensive knowledge of thermal effects on arthropods can be used to determine the range where the pest might develop or establish and to forecast the population’s growth potential and dynamics. Actually, a precise estimation of the pest’s population parameters is essential to develop a pest management programme [43].’ In Discussion.
Also, authors should explain how they have arrived to using solely the SSI model for their development predictions, i.e., the estimated thresholds may not quite equate to the biological threshold above which this mite develops. Studies on the effects of temperature on insect and mite development have been criticized because analyses commonly use models that are considered standard to the field of investigation or are preferred for a particular taxonomic group (see https://doi.org/10.1093/aesa/saw098, or https://doi.org/10.1093/aesa/sax063). As a result, alternative models that could provide superior fits to experimental datasets may be overlooked. Other important criteria that should be considered for fitting nonlinear models to temperature-driven development rate data for insects include use of parameters that have biological relevance are close-to-linear, which means that the least squares estimators are close to being normally distributed and are unbiased, minimum variance estimators. In these situations, good initial parameter estimates can be obtained which promotes successful model convergence (see https://doi.org/10.1093/jee/toz320).
My final concern is that the authors are extrapolating the applicability of their results beyond what the design supports. These are only development data from sets of highly artificial constant temperatures, so the inference power of the paper is very limited, but authors do not acknowledge this detail at all and need to be more forthcoming. This is a critical limitation of the study, and the authors must concede and discuss this. It is well known that most of laboratory experiments are conducted under constant temperatures whereas in nature daily temperature fluctuations can be very wide. The interaction of cyclic temperatures with nonlinear development or life history parameters can introduce significant deviations from the parameters developed here. So, I am suggesting to the authors to tone-down the language a little and admit that there are still substantive uncertainties to be considered, including uncertainty as to how generalizable the results are to open field conditions. Some of the authors statement would for example be much stronger if they tie and compare their work to the body of literature that has built up from rearing insects and mites at a wide set of realistic fluctuating temperatures. This is not to diminish the data gathered in this study, they are of value. But it is important for the authors not to overgeneralize, and to warn the reader against doing so as well. Some suggestions might include but are not limited to J. Econ. Entomol. 113: 633-645., J. Econ. Entomol. 112: 1560-1574., J. Econ. Entomol. 112:1062-1072., etc. etc. Adding these details will improve the paper in my opinion.
- Added ‘The upper (TH) threshold temperatures were estimated at 35.2 and 36.9°C for egg-to-adult female and egg-to-adult male, respectively, suggesting that the hypothetical enzyme was half active and half inactive at these thresholds. This could be one of the explanations for the incomplete development of the egg and larval stage at > 35° C.’ in Discussion
- To estimate the lower thermal threshold (t0), the optimal temperature (Topt) and the lethal maximum temperature (Tmax), there are several models are used such as Analytis, Briese-1 and -2, Hansen, Kontodimas, Lactin, Logan-3, Performance, Regniere and Thermodynamic SSI model (Ikemoto and Takai 2000; Kontodimas et al. 2004; Kheradmand et al. 2007; Sanchez-Ramos et al. 2015; Ikemoto and Kiritani 2019). However, from an applied point of view, to estimate the number of generations per year is indispensable, so we have to know the thermal constant (thermal summation, K) of pests and predators. Thermal constant is only calculated using linear model. Of course, most models enable to the estimation of two or three parameters of them, but the only model that can estimate all four is Thermodynamic SSI (Campbell et al. 1974; Ikemoto and Takai 2000; Ikemoto and Kiritani 2019). In this study, therefore, development rates (calculated as 1/development time) at different constant temperatures were used SSI model.
- Added ‘It is vital to construct a precise predictive model for adult emergence, and a forecast strategy that can assist as a critical component of an integrated pest management system wherein it enables decision making and improves control efficacy. Predicting the precise seasonal occurrence of agricultural insect/mite pests including kankitus is significant for scheduling, sampling, and the selection of control tactics. Climatic factors in general have been shown to play substantial roles in insect life; among such climatic factors, temperature has the utmost influence on population dynamics and the timing of biological events of insect and mite species. The development of mites occurs within narrow temperature ranges that vary between different mite species, and is sensitive to temperature changes. There are still substantive uncertainties to be considered, including uncertainty as to how generalizable the results are to open field conditions. The survival rates, development rates, reproductive capacities and population growth of spider mites under fluctuating temperatures are disparate from those under constant temperatures [13, 44, 45]. Henceforth, fluctuating temperatures should take into account wisely to predict the population dynamics of spider mites in nature.’ in Discussion
Round 2
Reviewer 3 Report
Authors have done a nice job addressing all of my original comments and those of other reviewers. I have no further suggestion for improvement. Thank you.
Author Response
Thank you for your approval.
Our paper has been reviewed in English by American ecologist working at University of Nevada.